# The Mechanisms of Sodium Chloride Stress Mitigation by Salt-Tolerant Plant Growth Promoting Rhizobacteria in Wheat

Zhen Huang [1], Chen Wang [1,*], Qing Feng [1], Rey-May Liou [2], Ying-Feng Lin [2], Jinhua Qiao [1], Yaxin Lu [1] and Yuan Chang [1]

1    College of Environmental Science and Engineering, Qilu University of Technology (Shandong Academy of Sciences), Jinan 250353, China; huangzhenfwz@126.com (Z.H.); qingfeng@qlu.edu.cn (Q.F.); q15203582597@163.com (J.Q.); lu1382002@163.com (Y.L.); cy18963491435@163.com (Y.C.)
2    Department of Research, Development Centre of Ecological Engineering and Technology, Chia Nan University of Pharmacy and Science, Tainan 71710, Taiwan; mrmliou@mail.cnu.edu.tw (R.-M.L.); yflin@mail.cnu.edu.tw (Y.-F.L.)
*    Correspondence: shanqing123@126.com

**Abstract:** We investigated the growth-promoting mechanism of salt-tolerant plant growth promoting rhizobacteria (ST-PGPR) in wheat under sodium chloride (NaCl) stress by measuring the growth and physiological and biochemical responses of wheat plants inoculated with ST-PGPR under 0–400 mM NaCl. The results showed that ST-PGPR plays a significant role in the growth of wheat under NaCl stress. Under 300 mM NaCl, wheat plants inoculated with the three ST-PGPR strains increased in plant height, root length, dry weight, and fresh weight by 71.21%, 89.19%, 140.94%, and 36.31%, respectively, compared to the control group. The proline and soluble sugar contents of wheat inoculated with *Bacillus thuringiensis* increased by 38.8% and 21.4%, respectively. The average content of antioxidant enzymes increased by 13.89%, and compared with the control, in wheat inoculated with the three species of ST-PGPR, the average content of ethylene decreased 2.16-fold. In addition, a mathematical model based on the "interaction equation" revealed that the best results of mixed inoculation were due to the complementary strengths of the strains. The analysis of experimental phenomena and data revealed the mechanisms by which *Brevibacterium frigoritolerans*, *Bacillus thuringiensis*, and *Bacillus velezensis* alleviate NaCl stress in wheat: (1) by lowering of osmotic stress, oxidative stress, and ethylene stress in wheat and (2) by using root secretions to provide substances needed for wheat. This study provides a new approach for the comprehensive understanding and evaluation of ST-PGPR as a biological inoculant for crops under salt stress.

**Keywords:** ST-PGPR; NaCl stress; salt tolerance mechanism; mathematical model; microbial agents

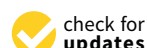



## 1. Introduction

Soil salinization is a worldwide ecological problem that severely limits the growth and yield of crops [1]. It is estimated that 954 million hectares of land worldwide are threatened by salt stress [2,3]. Currently, 11.3% of China's agricultural land is affected by salinization to varying degrees [4]. However, wheat, as the most important agricultural crop that provides approximately one-third of China's food source, is already severely affected by soil salinity [5].

NaCl stress is considered to be one of the main factors limiting wheat growth [6]. High salt (NaCl) concentrations disrupt the normal $Na^+/K^+$ ratio, resulting in impaired osmotic phenomena in plants [7,8]. Moreover, the presence of $Na^+$ and $Cl^-$ can cause ion toxicity and nutrient deficiency, which affects the normal physiological activities of cells [9,10]. In addition to osmotic stress and ion toxicity, another devastating effect of salt stress on plants is oxidative damage, which is caused by the accumulation of reactive oxygen species (ROS) brought on by osmotic stress and ion toxicity, leading to the hindrance of secondary metabolism and the occurrence of oxidative stress [9,11]. Superoxide ($O_2^-$), hydroxyl

radicals (OH$^-$), and hydrogen peroxide (H$_2$O$_2$) are the main toxic ROS in plants [12,13]. Excessive accumulation leads to membrane lipid peroxidation reactions that damage cell membranes, causing irreversible metabolism, structural damage, and ultimately, cell death [14]. Plants alleviate salt stress in the natural environment mainly through the following mechanisms: (1) accumulation of osmoregulatory substances, such as proline and soluble sugars, to maintain cellular osmotic potential; (2) separation of salt ions from vacuoles; and (3) regulation of the antioxidant defense system and removal of toxic ROS [15,16].

Salt-tolerant plant growth promoting rhizobacteria (ST-PGPR) alleviate salt stress in plants through direct and indirect regulatory mechanisms [17]. Direct regulation occurs through the production of phytohormone signals that aid plant growth and development, such as indole-3-acetic acid (IAA), and by other chemical processes, such as leaching phosphorus, fixing nitrogen, and increasing nutrient uptake. Indirect regulation is brought about by an increase in the plant's defense system and changes in the metabolic pathways that promote the accumulation of antioxidants and osmoregulatory substances [18,19]. In recent years, an increasing number of ST-PGPRs have been found to enhance IAA resistance in plants under salt stress and alleviate the stress caused by excessive ethylene levels [20,21]. However, at present, most studies on the mechanism of ST-PGPR in alleviating salt stress in wheat have been conducted with single strains rather than mixed strains of inoculants, and the specific material pathways for ST-PGPR that affect IAA and ethylene changes in wheat are also unclear. We hypothesized that the material pathways in single and mixed strains of bacteria to alleviate salt stress in wheat may be related to the root secretions.

In a recent study [22], it was found that three ST-PGPR strains selected from saline soil promote growth in wheat under NaCl stress. To further explore the mechanism by which the three ST-PGPR strains promote wheat growth under NaCl stress, inoculation experiments were conducted with three ST-PGPR strains (*Brevibacterium frigoritolerans*, *Bacillus thuringiensis*, and *Bacillus velezensis*) under different NaCl concentrations. A pot experiment with an orthogonal design was used to reveal the mechanism by which ST-PGPR promotes the growth of wheat in saline soil. The objectives were as follows: (1) to analyze the effect of inoculation with ST-PGPR on the production of osmoregulatory substances and the synthesis of antioxidant enzymes in wheat; (2) to estimate the specific material pathways by which inoculation with ST-PGPR affects changes in IAA and ethylene in wheat; (3) to investigate why the mixed inoculation of the three species of ST-PGPR has a stronger growth-promoting effect on wheat than any single inoculation; and (4) to clarify the mechanism of ST-PGPR in improving the salt tolerance of wheat.

## 2. Materials and Methods

### 2.1. Plant and Microbial Strains

*Triticum aestivum* L. (wheat) seeds were purchased from the agricultural seed market in Shandong Province, China. The three strains of bacteria (*Brevibacterium frigoritolerans*, *Bacillus velezensis*, and *Bacillus thuringiensis*) have the ability to tolerate salt and promote plant growth, as demonstrated in the previous research of this project [22].

### 2.2. Experimental Design

Three parallel sets of pot analysis experiments (5 × 5) were designed and implemented. Thus, there were 75 experimental units under a completely orthogonal design, including five levels of salt treatments (0, 100, 200, 300, and 400 mM NaCl) and five levels of bacterial treatments (no bacteria, CK; *Brevibacterium frigoritolerans*, *Bf*; *Bacillus velezensis*, *Bv*; *Bacillus thuringiensis*, *Bt*; and a 1:1:1 mix of the three strains).

Before seed germination, wheat seeds with full grains were selected for sterilization [17]. The seeds were evenly spread, with 10 seeds in each pot (8.0 cm diameter and 9.0 cm height). In order to avoid the influence of other factors, the test soil was sterilized three times using high temperature and pressure (121 °C, 30 min).

The three strains were cultured in nutrient broth medium (1000 mL distilled water, 15 g peptone, 5 g beef extract, 1.8 g sodium chloride, and 5 g glucose) and cultured in a constant temperature shaker (120 rpm, 25 °C) for 48 h. The cultures were then centrifuged at 4000 rpm for 5 min. The sediment was mixed well with sterile diluted Hoagland nutrient solution to obtain a bacterial inoculum of approximately $2 \times 10^9$ CFU·mL$^{-1}$ for the inoculation experiments. After the wheat seeds were sown, the experimental group was evenly inoculated with a bacterial suspension around the seeds, and the control groups were inoculated with 20 mL of sterile water. Inoculation was carried out continuously for 7 d (20 mL per day). During this period, five seedlings with similar growth were left in each pot. At this point, the inoculation experiment was stopped, and NaCl salinity treatment commenced. The study was conducted in a simulated greenhouse, with controlled temperatures of 25–30 °C, and wheat plants were harvested after 28 d of growth.

### 2.3. Transmission Electron Microscopy (TEM) Analysis

Fresh wheat in pots was selected for the TEM analysis, 1–2 mm$^3$ of root tissue was cut with a razor blade, and the collected samples were fixed with 2.5% glutaraldehyde solution and then stored for 12 h at 3–5 °C in the refrigerator. The samples were rinsed three times with 0.1 M of phosphate buffer (pH = 7) for 15 min each time, and then the samples were fixed with 1% osmium acid solution for 1–2 h. The samples were treated with 100% ethanol for 20 min; finally, they were over-treated into pure acetone for 20 min. The treated samples were embedded and heated at 70 °C for 12 h to obtain the embedded samples. Finally, the samples were sliced and observed in TEM.

### 2.4. Growth Status Assessment on Wheat

2.4.1. Analysis of Substances Related to Osmotic Stress and Measurement of Growth Parameters

Proline (Pro) was detected by the acidic ninhydrin colorimetric method. An amount of 5 mL of 3% ($w/v$) sulfosalicylic acid was added in 0.2 g of fresh leaves, and then grinded in an ice bath and filtered with funnel. Then, 2 mL of ninhydrin and 2 mL of glacial acetic acid was added into the filtrate, and heated in a water bath at 100 °C for 40 min. The proline of samples was extracted with 5 mL of toluene and finally measured the absorbance at 520 nm. Soluble sugar content was measured using the anthrone colorimetry method [23,24]. After wheat was incubated in a simulated greenhouse (25–30 °C) for 28 days, the whole seedlings were removed from the pots, the height and root length of each plant were measured, and the fresh and dry weights (dried at 60 °C for 24 h) were determined using an electronic balance [25].

2.4.2. Analysis of Plant Antioxidant Systems

The method of Güneş et al. was modified for the determination of peroxidase (POD) activity, 5 mL of 20 mmol/L $KH_2PO_4$ was added to 0.2 g of wheat leaves for ice bath grinding, centrifuged at 8000 rpm for 10 min, the supernatant taken, and 2% $H_2O_2$ and 0.05 mol/L guaiacol added in the reaction system, which was measured by monitoring the change in absorbance at 470 nm [26]. Superoxide dismutase (SOD) activity was determined using the nitroblue tetrazolium chloride method, which was determined based on the absorbance of NBT at a wavelength of 560 nm [27]. Catalase (CAT) activity was measured as the change in $H_2O_2$ content in the reaction system [28]. For peroxidation products, malondialdehyde (MDA) in the plants was measured as previously described. An amount of 0.2 g of leaves were taken, 5 mL of 5% trichloroacetic acid (TCA) was added to the homogenate, and then 1 mL of 0.5% ($w/v$) thiobarbituric acid (TBA) was added. The water bath was heated (100 °C) for 20 min and the absorbance was measured at 532 nm and 600 nm to calculate the MDA content [23].

### 2.4.3. Detection of Amino Acids in Wheat Root Secretions

The soil attached to the roots of wheat was extracted, and 95% ethanol was added at a volume ratio of 1:1. This mixture was shaken, centrifuged and the supernatant was evaporated to dryness, and the volume was fixed with 50% ethanol to 1/10 of the pre-concentration volume and reserved as mother liquor. The amino acid content was detected with amino acid analyzer (Essentia LC-16AAA, Shimadzu, Suzhou, China), and the chromatographic column was AccQ-Tag. The settings of the Tag amino acid analysis column were as follows: flow rate, 1 mL·min$^{-1}$; column temperature, 37 °C; and detection wavelength, 248 nm.

### 2.4.4. Determination of Ethylene Content

The wheat seeds were sterilized and then incubated with the bacterial suspension for 2–4 h. NaCl solutions of different concentrations (0, 100, 200, 300, 400 mM) were added sequentially to each sealed vial. Three groups of bacterial suspensions, one group of mixed bacterial suspensions and one group of control experiments were performed according to this method. After one week, the headspace volume in the bottle was collected, and the ethylene content was detected using a gas chromatograph (GC-2014C, Shimadzu, Suzhou, China) [29].

### 2.5. Detection of IAA-Producing Ability and ACC (1-Aminocyclopropane-1-Carboxylate) Deaminase Activity of the Bacterial Strains

To determine the IAA-producing ability of the strain, 2 mL of bacterial solution was added and inoculated for 48 h in 8 mL of nutrient broth medium containing tryptophan (100 mg·L$^{-1}$), after which 2 mL of supernatant was added to 4 mL of Salkowski chromogenic solution. The mixture was then placed in the dark for 0.5 h, and the absorbance was measured at 540 nm [30]. The determination method of ACC deaminase activity was as follows: The effects of 0, 100, 200, 300, 400 mM NaCl on the growth of the strain were studied on DF (Dworkin and Foster) medium supplemented with 2 g/L ACC, and the absorbance of the samples was measured at a wavelength of 540 nm with 0 mM medium as a control. Because the cleavage of ACC deaminase produces $\alpha$-ketobutyrate and ammonia, the content of $\alpha$-ketobutyrate produced during cleavage can be combined with the standard curve of $\alpha$-ketobutyrate to determine the activity of ACC deaminase [31].

### 2.6. Statistical Analysis

### 2.6.1. Principal Component Analysis (PCA)

To integrate all the results of the analysis, this study used principal component analysis (PCA) to explain the relationship between the treatment factors and the actual results and to identify the key factors of ST-PGPR affecting the growth of wheat in salinization areas.

### 2.6.2. Growth-Promoting Model of Compound Bacteria

In this study, a model inspired by the "interaction equation" of [32] was designed to explore the growth-promoting effects of the three ST-PGPR agents under NaCl concentration of 0–300 mM. The SPSS19.0 software program was used to perform analysis of variance and linear regression analysis of the data. The conceptual model was constructed as follows:

$$\sum_{j=1}^{m} y_j = k \sum_{i=1}^{n} (b_i x_i) + l \tag{1}$$

In the present experiment, let

$$X = \sum_{i=1}^{n} (b_i x_i) \tag{2}$$

$$Y = \sum_{j=1}^{m} y_j \tag{3}$$

where $y_j$ represents the wheat characteristics (SOD, POD, CAT, Pro, and soluble sugar); $x_i$, the growth-promoting properties of strains (IAA and ACC deaminase); $b_i$, the correlation

coefficient (the sum of r-values of partial correlation analysis of a certain growth-promoting characteristic to all plant characteristics) (Table S1); *k*, the linear coefficient; and *l*, the linear intercept.

In this study, the data were de-dimensionalized using statistical standardization in order to avoid the unit distorting the relationship of the variables. The data obtained from these calculations were substituted into the Origin drawing software (performed on Origin 2021 software), and the general linear model of the growth-promoting effect of the compound bacterial agent was fitted.

## 3. Results and Discussion

### 3.1. TEM (Transmission Electron Microscope) Analysis

The images of wheat roots obtained through TEM show that there were obvious differences in the wheat roots before and after inoculation with ST-PGPR, which indicated that ST-PGPR might have some connection with the roots of wheat (Figure 1). Studies by some scholars have shown that the microenvironment of plant rhizosphere soil will have a certain guiding effect on microorganisms [21]. Therefore, this study speculates that the mechanism of ST-PGPR regulating wheat growth may be related to wheat root exudates.

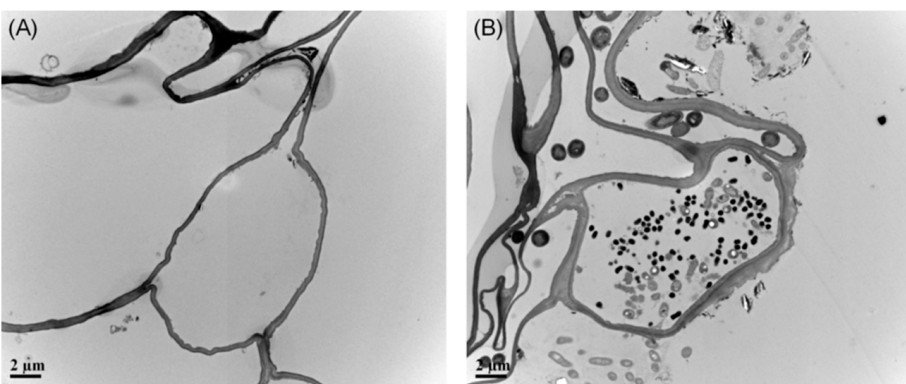

**Figure 1.** Morphology of wheat roots under transmission electron microscope: (**A**) wheat roots not inoculated with salt-tolerant plant growth promoting rhizobacteria (ST-PGPR) and (**B**) wheat roots inoculated with ST-PGPR.

### 3.2. Analysis of Wheat Growth Status

#### 3.2.1. Growth Characteristics

At 0–400 mM NaCl, the growth status of wheat deteriorated with increasing NaCl concentration, as evidenced by the decrease in four indicators: plant height, root length, dry weight, and fresh weight (Figure 2). However, the growth characteristics of wheat were significantly improved after inoculation with ST-PGPR. The best growth of wheat was achieved by the mixed inoculation of Bf, Bv, and Bt, and compared to the control, mixed inoculation condition led to an average increase of 71.21%, 89.19%, 140.94%, and 36.31% in the plant height, root length, dry weight, and fresh weight, respectively, and significantly ($p < 0.05$) improved the growth status of wheat (Table S2), which indicates that there is a synergy between the three ST-PGPR strains. Similar results were also found in a study by [33], where ST-PGPR had a significant promoting effect on the growth of wheat under NaCl stress. The mechanism of the synergistic effect of the three ST-PGPR strains are discussed in Section 3.5.

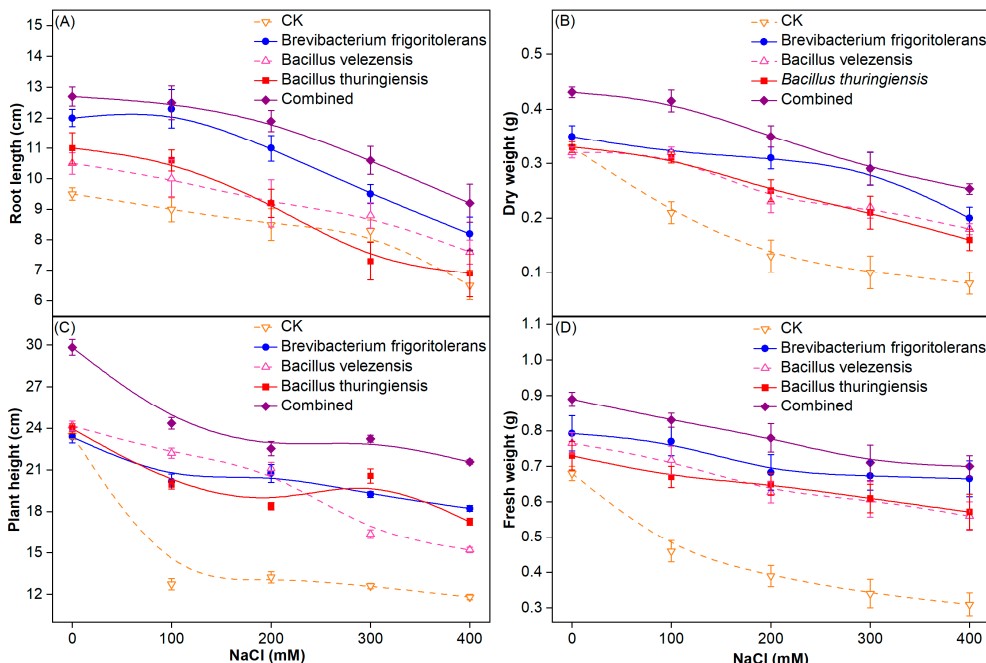

**Figure 2.** Effect of salt-tolerant plant growth promoting rhizobacteria (ST-PGPR) on wheat morphological characteristics under NaCl stress: (**A**) wheat root length, (**B**) wheat dry weight, (**C**) wheat plant height, and (**D**) wheat fresh weight.

### 3.2.2. Regulation of Osmotic Pressures

The two main substances that regulate osmolarity in wheat are shown in Figure 3. Under 0 mM NaCl, the differences in proline and soluble sugar contents between the control and experimental groups were minimal, indicating that, in the absence of stress, wheat can maintain normal osmotic pressure and microbial involvement does not have much effect. When the plants were subjected to salt stress (increase in NaCl concentration), there was a significant ($p < 0.05$) difference between the control and experimental groups as the NaCl concentration increased (Table S3). All three strains had a regulatory effect on the osmotic pressure of wheat compared with the control. After the NaCl concentration reached 300 mM, the proline and soluble sugar contents began to show a decreasing trend, indicating that ST-PGPR activity also began to decrease under high salt stress, in parallel with the trend of the growth characteristics of wheat. Wheat inoculated with *Bacillus thuringiensis* increased proline and soluble sugar levels by 38.8% and 21.4%, respectively, under 300 mM NaCl. *B. thuringiensis* showed a more pronounced regulatory effect than the other two strains, suggesting that it may be the dominant factor in osmoregulation. The three strains of ST-PGPR enhanced the accumulation of soluble sugars in wheat, such as glucose and sucrose, which is related to stomatal movement and leaf photosynthesis [34]. Moreover, the increase in soluble sugars had an effect on ROS, which is the main substance of oxidative stress [35]. In this study, ST-PGPR increased the content of soluble sugars, and the trend of soluble sugars (Figure 3A) was similar to that of antioxidant enzymes (Figure 4). Therefore, the osmoregulation and antioxidant regulation in wheat may be a system of mutual influence under the effect of ST-PGPR.

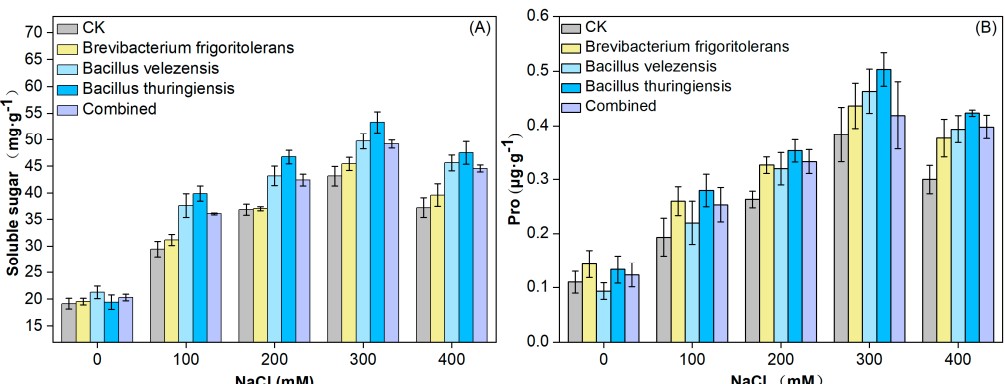

**Figure 3.** Effect of salt-tolerant plant growth promoting rhizobacterial (ST-PGPR) on wheat osmotic substances under NaCl stress: (**A**) soluble sugar and (**B**) proline (Pro).

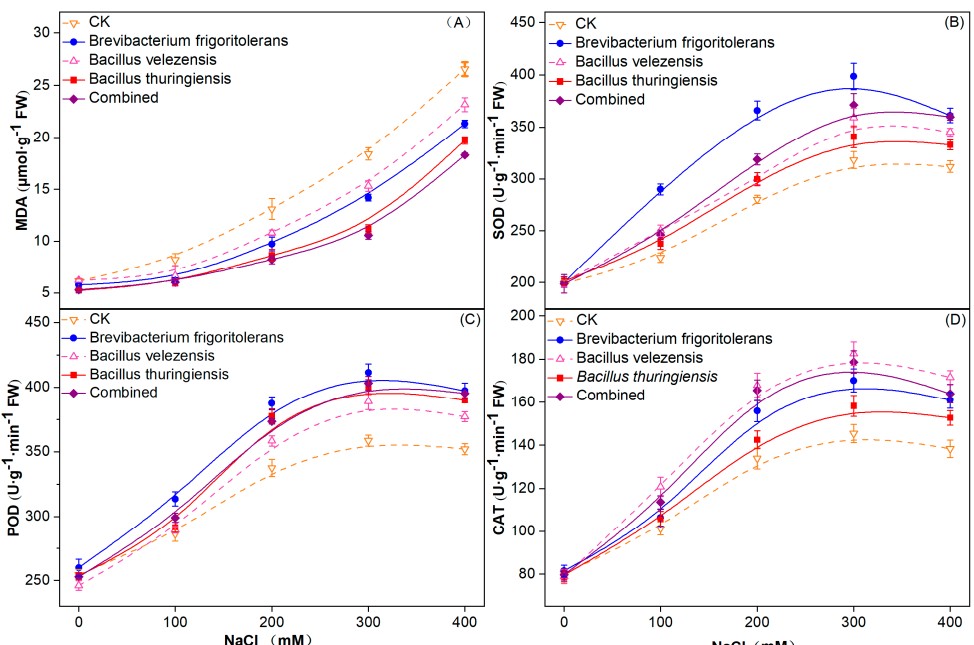

**Figure 4.** Effects of salt-tolerant plant growth promoting rhizobacteria (ST-PGPR) on (**A**) malondialdehyde (MDA) contents, (**B**) superoxide dismutase (SOD), (**C**) peroxidase (POD), and (**D**) catalase (CAT) activity in wheat under NaCl stress.

### 3.2.3. Regulation of the Antioxidant System

Consistent with the results of Section 3.2.2, high salt stress caused the activity of ST-PGPR to start decreasing when the NaCl concentration exceeded 300 mM, as shown in Figure 4, and the changes in antioxidant enzymes (SOD, POD, and CAT) followed this trend. Under 0 mM NaCl, inoculation with ST-PGPR had no significant effect on the changes in SOD, POD, and CAT activities in wheat compared to the control group. However, in the condition of 300 mM NaCl, *Brevibacterium frigoritolerans*, *B. velezensis*, and *B. thuringiensis* caused an increase in SOD activity by 23.98%, 20.10%, and 6.47%, respectively. POD activity also increased by 12.89%, 7.73%, and 10.50%, and CAT activity by 14.71%, 20.32%, and 8.23%, respectively. The average increase after inoculation with ST-PGPR was 13.89%. For the salinity-stressed wheat, antioxidant enzyme activity was significantly different between untreated and strains-treated wheat seeds (Table S4, $p < 0.05$). In contrast to the increase in the activity of antioxidant enzymes, wheat inoculated with ST-PGPR showed different degrees of decrease in the MDA content (Figure 4A). MDA is the end product of membrane lipid peroxidation, which is an important indicator of oxidative stress in plants.

The decrease in MDA content and the increase in antioxidant enzyme content indicate that ST-PGPR effectively maintained the antioxidant system of wheat.

Under NaCl stress, the common ROS radical in plants is $O_2^-$. SOD is the main defense enzyme against $O_2^-$ as it is able to catalyze transformation of $O_2^-$ into $H_2O_2$, which is then broken down into $H_2O$ and $O_2$ through the enzymatic activity of POD and CAT. As a result, oxidative stress is alleviated [36,37]. Previous studies [38,39] have shown that the activities of beneficial microorganisms in the soil can increase the activity of CAT and POD, thereby slowing down membrane lipid peroxidation in host plants. In an earlier study [40], similar changes were reported in the sensitivity of antioxidant enzymes to salinity. The results of the present study show that the mixed strains exhibited stable and strong regulation of all three antioxidant enzymes, which may be due to the synergistic effect between the strains. In addition, *B. thuringiensis* played a dominant role in the regulation of soluble sugars. The greatest increases in SOD and POD were observed in the *Brevibacterium frigoritolerans* treatment; thus, this species appears to be the dominant factor regulating antioxidant enzymes. It would appear that the reason for the best effect of mixed inoculation is the complementary advantages between the strains.

### 3.3. Material Pathways of IAA Content Changes in Wheat Are Regulated by ST-PGPR

IAA is one of the most important growth factors in plants and can affect cell division and root growth. When plants are subjected to salt stress, IAA content decreases, thus affecting plant growth. The IAA synthesized by ST-PGPR cultured in a medium containing L-tryptophan under different NaCl stress conditions is shown in Figure 5C. When the NaCl concentration reached 300 mM, the highest IAA production (57.19 mg·$L^{-1}$) was obtained in the ST-PGPR treatment. In contrast, no IAA component was detected in the metabolites of ST-PGPR when tryptophan was not added to the medium (as shown in Table S5). This indicates that the three ST-PGPR strains utilized the L-tryptophan pathway to synthesize IAA. Interestingly, we detected L-tryptophan in the assay of wheat root secretions (Table 1) and confirmed that ST-PGPR can colonize wheat roots in a preliminary TEM experiment. Therefore, this study elucidates for the first time the material pathways by which these three ST-PGPR strains regulate IAA changes in wheat by using wheat root tryptophan to synthesize IAA, thereby alleviating the inhibitory effect of NaCl stress on IAA in wheat. This finding is of great practical importance given that it has been reported that IAA produced by ST-PGPR can promote plant growth by promoting IAA uptake and improving tolerance to environmental stress [41]. IAA dissolved in aqueous solution and protonated can passively diffuse into plant cells without specific transporter proteins [42]. In the present study, under the same tryptophan substrate concentration, the IAA production of ST-PGPR showed an increasing trend with the increase in NaCl concentration and reached a maximum at 300 mM, indicating that stress of NaCl stimulated the stress response of ST-PGPR, accelerated the acquisition of nutrients and improved the utilization of tryptophan.

### 3.4. ST-PGPR Regulates the Material Pathways of Ethylene Content Changes in Wheat

The ACC deaminase activity of ST-PGPR under different NaCl stresses is shown in Figure 5A. The maximum ACC deaminase activity was reached at 300 mM NaCl, and the ACC deaminase activity of the mixture of strains reached 21.53 μmol α-K·$h^{-1}$·$mg^{-1}$, which is 1.59 times greater than the value in the *Bacillus velezensis* treatment (13.52 μmol α-KB·$h^{-1}$·$mg^{-1}$), which is consistent with the results shown in Figure 5B, where the mixed strains had a significant effect on the reduction of ethylene content under NaCl stress. The ethylene level in the CK group without ST-PGPR inoculation increased 2.94-fold with increasing NaCl concentration (0 mM, 7.22 ng·$mL^{-1}$·$h^{-1}$; 400 mM, 21.22 ng·$mL^{-1}$·$h^{-1}$). In contrast, in wheat inoculated with ST-PGPR, ethylene levels were consistently in the range of 6.89–10.74 ng·$mL^{-1}$·$h^{-1}$, with an average reduction of 2.16-fold compared to the CK group. Ethylene is an endogenous hormone produced by all higher plants and plays an important regulatory role in plant growth and development [43,44]. However, under

salt stress—in particular, elevated levels of NaCl—plant ethylene levels increase significantly, and the resulting ethylene stress limits growth [45]. ACC (1-amino-cyclopropane-1-carboxylic acid) is a synthetic precursor of ethylene, and ACC deaminase can break down ACC into $\alpha$-ketobutyric acid and $NH_4^+$, thus reducing ethylene levels in wheat [46]. Therefore, the results of this study show that even at a high salinity of 400 mM, ST-PGPR still had a strong ability to degrade ACC, even though ST-PGPR activity had decreased. These results demonstrate that the ability of ST-PGPR to alleviate ethylene stress is a more dominant factor than is osmotic or oxidative stress, and that ethylene is the main factor affecting the growth of wheat when subjected to NaCl stress.

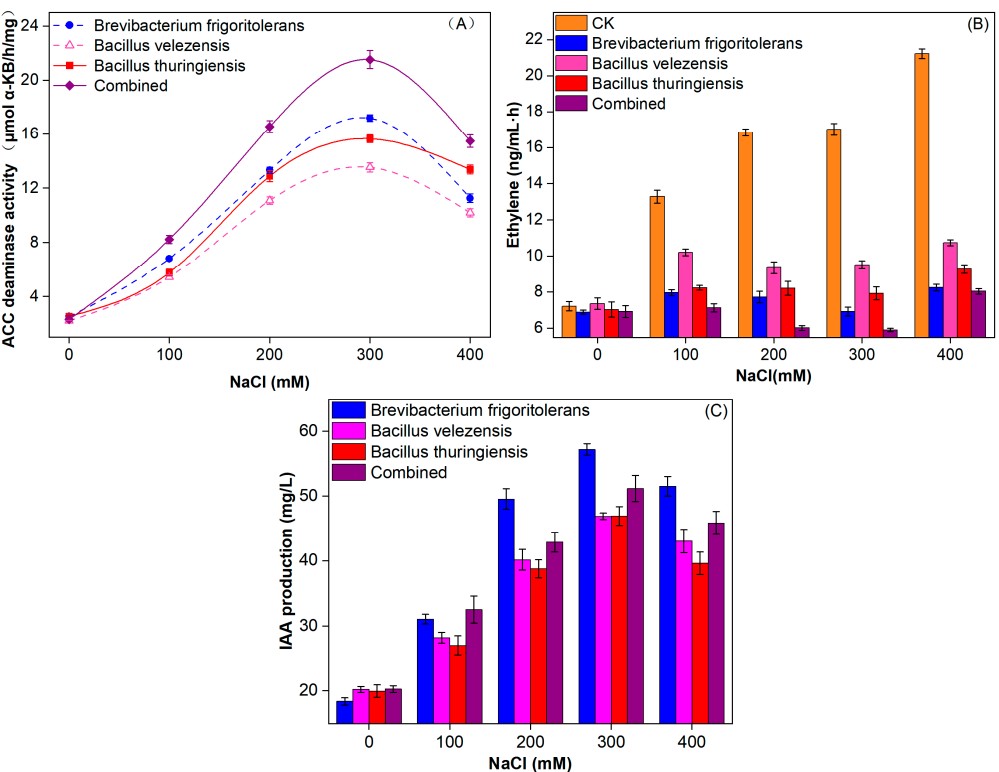

**Figure 5.** Effects of NaCl stress on (**A**) ACC deaminase activity of salt-tolerant plant growth promoting rhizobacteria (ST-PGPR), (**B**) ethylene levels in wheat, and (**C**) indole-3-acetic-acid (IAA) production of ST-PGPR.

ST-PGPR synthesized IAA using tryptophan from wheat roots, and IAA produced by bacteria was found to activate the activity of plant ACC synthase in a study by [45]; all plants associated with IAA-synthesizing bacteria had higher levels of ACC and ethylene. Thus, IAA produced by ST-PGPR fully inhibits plant growth when other mechanisms are not involved. However, this is not the actual result because when ethylene levels are elevated, the increased ethylene causes the plant to develop a feedback mechanism that inhibits IAA signaling, which limits the transcription of ACC synthase and reduces ethylene production. We posit that the three ST-PGPR strains in this study utilized these two interrelated mechanisms to produce IAA, which promoted the growth of wheat while providing good control of ethylene levels in vivo.

**Table 1.** Effect of NaCl on amino acid excreted by wheat root.

| NaCl Concentration (mM) | | 0 | 100 | 200 | 300 | 400 |
|---|---|---|---|---|---|---|
| Content of amino acid excreted by wheat root (mol·L$^{-1}$) | Pro | 0.82 | 1.14 | 0.96 | 0.33 | 1.02 |
| | Met | 1.22 | 2.20 | 0.98 | 1.31 | 0.99 |
| | Trp | 0.35 | 5.52 | 2.81 | 1.32 | 0.61 |
| | Glu | 1.01 | 5.33 | 4.12 | 1.34 | - |
| | Asp | 0.42 | 3.11 | 1.85 | 0.81 | - |
| | Phe | 1.88 | 1.91 | 0.97 | 1.01 | 1.69 |
| | Leu | 4.22 | 3.95 | 2.77 | 2.36 | 3.17 |
| | Cys | - | 1.02 | 2.19 | 1.63 | 0.86 |
| | Val | 6.39 | 3.47 | 5.81 | 3.59 | 5.66 |
| | Ala | 7.21 | 9.43 | 5.29 | 3.61 | 5.22 |
| | Tyr | 1.06 | 2.37 | 2.53 | 1.67 | 1.21 |
| | Thr | 3.69 | 5.28 | 2.11 | 0.72 | - |
| | Ser | 2.02 | 5.17 | 1.06 | 3.37 | 3.41 |
| | Gly | 2.35 | 8.66 | 1.42 | 2.53 | 6.17 |

Note: "-", Not detected.

### 3.5. Statistical Analysis of the Mechanism of Salt Tolerance Enhancement in Wheat by Three Strains of ST-PGPR

#### 3.5.1. Effects of ST-PGPR on Key Factors Underlying Plant Growth

PC1 and PC2 accounted for 81.8% of the total variability in the data (Figure 6A). The loading of POD, CAT, PRO, MDA, and root length in PC1 (58.6%) were the major contributors, whereas PC2 (22.7%) was greatly affected by ethylene, dry weight, and fresh weight. Under 100 mM NaCl, PC1 showed that the root length of plants in the co-inoculation treatment had a higher score, indicating that co-inoculation may have a greater impact on the growth characteristics of wheat. Under 400 mM NaCl, the negative MDA scores of wheat plants that were co-inoculated were higher in PC1, whereas the positive scores of dry weight and fresh weight of plants that were co-inoculated were higher in PC2, indicating that co-inoculation may relieve salt stress in wheat. In addition, the highest negative values of ethylene were found in plants in the CK group in PC2 under 400 mM NaCl, once again confirming that ethylene is the most important factor affecting wheat growth under NaCl stress.

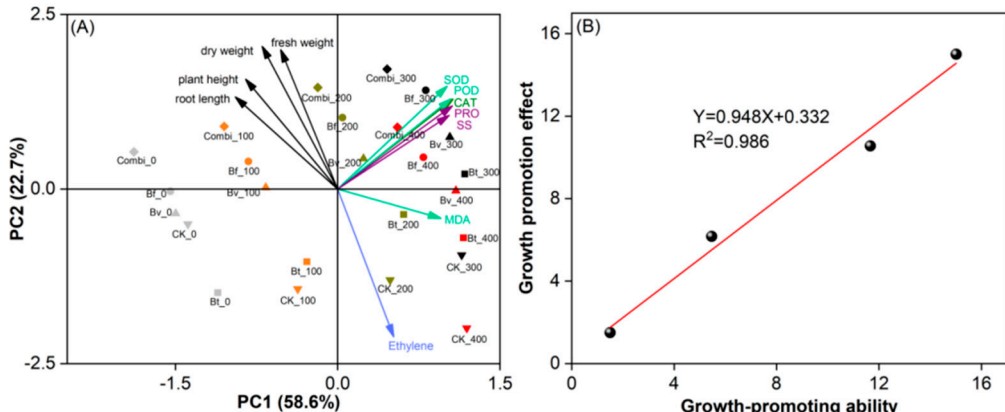

**Figure 6.** Effect of salt-tolerant plant growth promoting rhizobacteria (ST-PGPR) on the growth-promoting mechanism of wheat was investigated by using (**A**) principal component analysis and (**B**) linear regression analysis.

#### 3.5.2. The General Linear Model of ST-PGPR's Growth-Promoting Ability and Growth-Promoting Effect

The experimental results show that under NaCl stress of 0–300 mM, although the three ST-PGPR composite inoculants were not the dominant factor in the accumulation of

antioxidant enzymes and soluble osmotic substances, the composite inoculants result in the strongest growth-promoting effects in wheat. Therefore, we speculate that the reason why mixed inoculation outperforms single inoculation may be the synergistic effect of the three microorganisms, and this synergistic effect is manifested by the complementary advantages of different indicators.

As shown in Figure 6B ($Y = 0.948X + 0.332$), the critical value of the correlation coefficient, $r_{min} = 0.811$, was found when $\alpha = 0.05$ and $n = 4$, so $r = 0.997 > r_{min}$, indicating that the growth promotion ability of the bacterial inoculant has a high degree of linear correlation with the growth promotion effect. In other words, the equation exhibited a significant linear relationship. This indicated that the constructed general linear model was significant; that is, in the suitable salt concentration range, the growth-promoting effect of the complex bacterial agent on wheat increased with increasing salt concentration. $Y = 0.948X + 0.332$, when $X = 0$, the intercept is positive, indicating that ST-PGPR has no effect; that is, when it is not under NaCl stress, even without the participation of ST-PGPR, wheat itself has the ability to grow. This conforms to the natural law that plants can grow normally in the natural environment. The slope indicates the growth-promoting ability of bacterial agents. Under the same concentration of bacterial agent, the greater the slope, the stronger the growth-promoting effect of the bacterial agent, which can be used for comparisons between different microorganisms.

In this study, the effects of three ST-PGPR single inoculations were superimposed to obtain a linear model of the growth-promoting characteristics and effects of the compound bacterial agent. To verify the model, we substituted the experimental data of three kinds of ST-PGPR mixed inoculations for testing and found that the average degree of fit reached 89.71% (Table S6), which proves that the constructed general linear model has practical significance. This outcome also confirmed the hypothesis we put forward in the earlier stage; that is, the reason that mixed inoculation is better than single inoculation is the synergy of the three microorganisms, and this synergy is manifested as the complementary advantages of different indicators.

## 4. Conclusions

NaCl stress restricts the growth of wheat. In this study, three species of ST-PGPR—*Brevibacterium frigoritolerans*, *Bacillus thuringiensis*, and *Bacillus velezensis*—promoted the growth of wheat. Under salt-free conditions, the effect of ST-PGPR on wheat was not significant; however, ST-PGPR significantly improved the salt tolerance of wheat with increasing NaCl concentrations. ST-PGPR activity started to decrease when the NaCl concentration reached 300 mM. The main mechanisms through which ST-PGPR improves the salt tolerance of wheat are as follows: (1) by promoting the accumulation of proline and soluble sugars in wheat and alleviating osmotic stress caused by $K^+/Na^+$ imbalance; (2) by enhancing the antioxidant system of wheat and reducing the MDA content, thus slowing down the negative effects of membrane lipid peroxidation; (3) by using tryptophan from wheat root secretions to synthesize IAA to maintain wheat growth; and (4) by catabolizing ACC to $\alpha$-ketobutyric acid, and $NH_4^+$ substantially reducing ethylene stress. Moreover, the advantages of the three microorganisms complemented each other after co-inoculation, showing better results than a single inoculation. This study provides guidance for the use of ST-PGPR to promote the growth of crops in saltlands.

**Supplementary Materials:** The following are available online at https://www.mdpi.com/article/10.3390/agronomy12030543/s1, Table S1: Partial correlation analysis of growth-promoting characteristics and effects of salt-tolerant plant growth promoting rhizobacterial (ST-PGPR), Table S2: The effect of strains on wheat growth under different salt stress, Table S3: The effect of strains on the antioxidant system of wheat under different salt stresses, Table S4: The effect of strains on soluble sugars and proline in wheat under different salt stresses, Table S5: Capacity test of strain production of indole-3-acetic-acid (IAA), Table S6: Test of linear regression model.

**Author Contributions:** Z.H.: Conceptualization, Methodology, Validation, Investigation, Resources, Writing—original draft, Visualization; C.W.: Conceptualization, Validation, Writing—review and editing, Supervision, Funding acquisition; Q.F.: Writing—review and editing, Supervision; R.-M.L.: Resources, Visualization; Y.-F.L.: Resources, Visualization; J.Q.: Software, Visualization; Y.L.: Software, Visualization; Y.C.: Software, Visualization. All authors have read and agreed to the published version of the manuscript.

**Funding:** This research was funded by the Qilu University of Technology (Shandong Academy of Sciences) International Cooperative Research Special Fund Project (45040105), which we gratefully acknowledge.

**Institutional Review Board Statement:** Not applicable.

**Informed Consent Statement:** Not applicable.

**Data Availability Statement:** The data that support the findings of this study are available from the corresponding author upon reasonable request.

**Conflicts of Interest:** The authors declare no conflict of interest.

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
