# Peer review of "The Mechanisms of Sodium Chloride Stress Mitigation by Salt-Tolerant Plant Growth Promoting Rhizobacteria in Wheat"

_agronomy, doi:10.3390/agronomy12030543_

Round 1

Reviewer 1 Report

The manuscript titled “The mechanisms of NaCl stress mitigation by salt tolerant plant growth promoting rhizobacterial (ST-PGPR) in wheat” is an original paper. It expounds an innovative approach for the comprehensive understanding and evaluation of ST-PGPR as a biological inoculant for wheat under salt stress. The results reported are really interesting and increase the knowledge on the interaction between ST-PGPR and plants.

I have only one perplexity about Figure 2. Data reported in the manuscripts seems discordant with those in the figure. In particular, what do the percentage values refer to?

Minor revision:

Line 206 typos “Mm” maybe it is mM, please verify.

Line 250 Please put the name in italics

Line 282 please argue this lats phrase.

Line 310 Please delete

Reviewer 2 Report

The manuscript presented by Huang et al. aimed at identification of mechanisms of sodium chloride stress mitigation in wheat by salt-tolerant plant growth promoting bacteria. The manuscript is, however, of average importance, because it mostly recapitulates what was previously done by others. The authors themselves give us a view of this wide background by referring often to newest and older works, both in the introduction and discussion section (which was coupled to results here in this work). Taking this into account its worth to say, that the introduction provides a sufficient background, the research design is appropriate. However, the methods are not adequately described, and the conclusions are sometimes exaggerated and not supported by the results. English sounds standard. But there are many fragments that suffer from strange grammar.

All the comments are listed in the order of appearance of appropriate fragments of the text throughout the manuscript, irrelevant of their importance.

The title

“The mechanisms of NaCl stress mitigation by salt tolerant plant growth promoting rhizobacterial (ST-PGPR) in wheat” – in the title NaCl should be replaced with “sodium chloride”, while the abbreviation of plant-growth promoting bacteria should be omitted. Abbreviations should be explained the first time they appear and then used without explanation. The same applies to chemical formulas.

Throughout the text the authors use “PGP rhizobacterial” – it should be “rhizobacteria”, please correct throughout the manuscript.

Introduction

Here and in the rest of the manuscript body – the names of the species should be italicized.

L42: “osmotic metabolism” – rather osmotic phenomena

L62-65: please check if there’s no mistake in referencing the outcomes of work [20]

L77-83: How can we estimate pathways? Did the authors really investigated and found out why the mixed inoculation has stronger effect than single? Did the obtained results enabled to clarify the mechanism of improving salt tolerance, or maybe there are only some premises, fragmentary data and conclusions based on the previous studies and not with sufficient background in the obtained results?

Materials and methods

L86: The Latin name of the species and cultivar should be given.

Experimental design is very good.

L114-117: description of the TEM analysis is not sufficient, please give more details of how the root samples were collected and the whole preparation. “to observe the colonization of microorganisms” – it is the root that is colonized by microorganisms and not the opposite

The whole 2.4 paragraph should be more detailed, in every subparagraph the information were specific substances were measured – either in the plant of bacteria or secretions – should be given

L125: wheat was grown (not sown) for 28C?

L130-135: more details of the methods referenced in [22], [25], [26], and [27] should be given

L139: “…centrifuged and the supernatant was evaporated…”

L142-144: the description should be more detailed, and the name of the manufacturer should be given

L148-149: Was there any medium in the bottle? What was treated with NaCl? Awkward, please specify.

L153: bacterial strains

L153-160: ACC abbreviation should be expanded; more details of the method referenced in [30] should be given; “during ACC deaminase cleavage” – awkward, what cleaves what?

Results and discussion

“Colonization of microorganisms” – bacteria colonize roots, not the opposite. However, making a separate subsection for TEM analysis is unnecessary. Is the result new? I think the mode of colonization is well known.

L192-193: description details for Figure 1 should be given under the figure and are unnecessary in the text

L193: where can this “stable colonization” be seen?

L195-196: the information about Arabidopsis is irrelevant

L196-198: is it something new or unexpected?

Figure 1 itself lacks basic elements showing important and relevant plant structures and bacteria themselves; at the moment it is not self-explanatory; the Figure title sounds not scientific.

L206: mM not Mm

L213-216: if similar results were obtained also in this [34] and other works on wheat under salt stress, what is the novelty and strength of this work? “…will be discussed in subsequent studies” – further in the text or in the next article?

L222: the subtitle is awkward

L223: Substances are not presented in Figure 3. Please be specific.

L228, 232 and anywhere in the text: “saline wheat” – what is it?

L239-240: “…and the regulation of osmotic and oxidative stress is an interacting and linked system of ST-PGPR.” – what does it mean? Please specify.

L242: on the level of wheat osmotic substances

L245-259: “previous period?” – what do the authors mean? In the whole paragraph the authors use “content or concentration” when talking about enzymes, while the graphs mention activity – I think it is activity not the amount that should be used here. Please unify.

L270: catalyze transformation of O2- into H2O2

L279: should be toned down: “sugar content may also play a role in ROS-scavenging”

L287: “IAA synthesized” instead of “IAA content metabolized”

L296: IAA changes or IAA effect on wheat?

L302-307: what does it mean: “the same tryptophan substrate concentration” here? Table 1 shows that the concentration of secreted amino acids differs under different NaCl concentration; “tryptophan utilization in wheat roots” – but utilization by bacteria?

L310-311: some “inclusions”

L316: activity of the mixture of strains

L331-334: how can we say that ethylene stress is alleviated more profoundly then others

L336: “activate the activity” – please correct

L324: ethylene is not possessed but rather produced

L410: substantially reducing

Supplementary information

Table S2 was not mentioned anywhere in the manuscript
